# Global freshwater fish invasion linked to the presence of closely related species

Meng Xu [1,2,3] ✉, Shao-peng Li [4], Chunlong Liu [5], Pablo A. Tedesco[6], Jaimie T. A. Dick[7], Miao Fang [1,2,3], Hui Wei[1,2,3], Fandong Yu[1,2,3], Lu Shu[1,2,3], Xuejie Wang[1,2,3], Dangen Gu [1,2,3] ✉ & Xidong Mu [1,2,3] ✉

In the Anthropocene, non-native freshwater fish introductions and translocations have occurred extensively worldwide. However, their global distribution patterns and the factors influencing their establishment remain poorly understood. We analyze a comprehensive database of 14953 freshwater fish species across 3119 river basins and identify global hotspots for exotic and translocated non-native fishes. We show that both types of non-native fishes are more likely to occur when closely related to native fishes. This finding is consistent across measures of phylogenetic relatedness, biogeographical realms, and highly invaded countries, even after accounting for the influence of native diversity. This contradicts Darwin's naturalization hypothesis, suggesting that the presence of close relatives more often signifies suitable habitats than intensified competition, predicting the establishment of non-native fish species. Our study provides a comprehensive assessment of global non-native freshwater fish patterns and their phylogenetic correlates, laying the groundwork for understanding and predicting future fish invasions in freshwater ecosystems.

Freshwater biodiversity, which includes nearly 18,000 fish species constituting one-fourth of global vertebrates, is declining at unprecedented rates far greater than in other ecosystems[1,2]. Biological invasion has emerged as a prominent factor contributing to this decline[3–5]. Over 500 non-native freshwater fish species have been recorded as established worldwide, representing one of the most widely introduced taxonomic groups on Earth[6,7]. These established non-native fishes have led to a global homogenization of freshwater fish communities[1,8,9], resulting in significant ecological and socio-economic impacts on freshwater ecosystems[6,10,11]. Recognizing the global geographical patterns of these non-native species and comprehending the factors behind their prevalence in specific regions is essential for the effective management of these species and forecasting future invasions.

Global patterns and hotspots of non-native freshwater fishes have been assessed based on 1055 river basins[7,8]. However, this evaluation appears relatively limited, given that fish diversity has been well documented in over 3000 river basins worldwide[1,12]. Furthermore, understanding the origin of non-native fishes, whether they are introduced from other realms (exotic fishes), or translocated within a realm but different river basins (translocated fishes), is crucial to comprehend their invasions and the resulting homogenization of fish communities[1,13,14]. Nevertheless, there is still a deficiency in distinguishing translocated non-native fishes from exotic species and evaluating the difference in their global biogeographical patterns.

Comprehending the factors that contribute to the establishment of non-native fish species is challenging and not yet thoroughly

[1]Pearl River Fisheries Research Institute, Chinese Academy of Fishery Sciences, Guangzhou, China. [2]Key Laboratory of Prevention and Control for Aquatic Invasive Alien Species, Ministry of Agriculture and Rural Affairs, Guangzhou, China. [3]Key Laboratory of Alien Species and Ecological Security (CAFS), Chinese Academy of Fishery Sciences, Guangzhou, China. [4]Zhejiang Tiantong Forest Ecosystem National Observation and Research Station, School of Ecological and Environmental Sciences, East China Normal University, Shanghai, China. [5]The Key Laboratory of Mariculture, Ministry of Education, College of Fisheries, Ocean University of China, Qingdao, China. [6]UMR EDB, IRD 253, CNRS 5174, UPS, Université Toulouse 3 Paul Sabatier, Toulouse, France. [7]Institute for Global Food Security, School of Biological Sciences, Queen's University Belfast, Belfast, UK. ✉e-mail: xumeng@prfri.ac.cn; gudangen@163.com; muxd@prfri.ac.cn

understood. Many studies have attempted to address this issue by examining environmental adaptation[15], life-history strategies[16], human activity[7], propagule pressure[17] and functional traits[18]. Although these approaches provide valuable insights, they may pose challenges in reaching general conclusions and predictions for non-native fish species. The challenge primarily arises due to variations in ecological characteristics among species and habitats, coupled with the inherent difficulty in identifying and measuring functional traits[6,19]. Originating from Charles Darwin[20], the phylogenetic relatedness of non-native species to the natives of recipient regions has been considered the key to understanding and predicting the establishment of non-native species[21–23]. On the one hand, Darwin stated that non-native species phylogenetically close to native species would be more likely to establish successfully because they might share similar adaptations to the local environment with their native relatives, which was known as the pre-adaptation hypothesis[24]. On the other hand, Darwin also posited that non-native species phylogenetically distinct from the native species would tend to be more successful because they might share fewer natural enemies and face less competition with the native species, which was referred to as Darwin's naturalization hypothesis[23]. These two opposing hypotheses, which focus on phylogenetic relatedness between non-native species and recipient communities, have been coined as Darwin's naturalization conundrum[21,25]. While the use of phylogenetic relatedness to predict the fate of non-native species has been widely explored for plants[26–28], birds[29], and microbes[30], it has been rarely examined in fish communities[31]. Its effectiveness in predicting the establishment of non-native fish species and influencing their global geographical patterns remains largely unclear.

Building upon a philosophy akin to Darwin's naturalization conundrum, the diversity of fish in native communities can potentially yield contrasting outcomes for the establishment of non-native fish species. On the one hand, high native diversity may predict the success of non-native fishes, as the favorable environmental conditions sustaining a rich native species community should also benefit non-native species[7]. On the other hand, high native diversity is likely to impede the establishment of non-native fish species due to increased competition and fewer available ecological niches in the species-rich community[18,32]. Therefore, for a more explicit understanding of the role of phylogenetic relatedness in influencing the establishment of non-native fish species, it becomes essential to distinguish and account for the impact of native species richness. Furthermore, while taxonomic diversity metrics such as species richness may not adequately capture the diversity of ecological functions they support, comprehensive diversity metrics, such as the phylogenetic diversity of native communities, should be considered simultaneously[1,33]. By disentangling the comprehensive effects of taxonomic and phylogenetic diversities, and elucidating their potential indirect effects through phylogenetic relatedness, we can achieve a clearer understanding of how nonnative-native phylogenetic relationships affect the establishment of non-native fish species in global river basins.

Here, using the most comprehensive freshwater fish occurrence database, which includes 14,953 species across 3119 river basins in 143 countries, we explored the biogeographical patterns of non-native fishes and their phylogenetic correlates worldwide. We initially presented global biogeographical patterns for both exotic fishes introduced across countries and translocated fishes within those countries. Subsequently, we constructed a global phylogenetic tree of freshwater fish species (Fig. S1) to quantify the phylogenetic distances between each exotic or translocated fish species and native fishes in each river basin within a country. Thereby, we examined the relationship between nonnative-native phylogenetic relatedness and the likelihood of non-native fish occurrence at global, biogeographical realm, and country scales, respectively. We also extended the definition of exotic and translocated fish species from a country-level perspective to a biogeographical realm scale and reassessed the relationship between phylogenetic relatedness and occurrence. Additionally, we calculated the taxonomic and phylogenetic diversity of native fishes in each river basin and examined whether the relationship between phylogenetic relatedness and the occurrence of non-native fish species remains robust when considering the influence of native diversity. Our results indicate that non-native species phylogenetically close to native species are more likely to establish in freshwater fish community worldwide. The patterns and phylogenetic drivers of non-native freshwater fish species we revealed here will offer valuable insights for understanding, assessing, and predicting fish invasions in freshwater ecosystems in the Anthropocene.

## Results

### Global patterns of non-native freshwater fish species

Across the globe, out of 3119 river basins, 601 non-native fish species have successfully established in 1719 basins, representing 55.11% of the total. Among these, exotic fish species have established in 1518 basins, accounting for 50.69%, while translocated fishes have established in 603 basins, making up 19.33%. The top three river basins with the largest number of non-native fish species are the Colorado, Mississippi, and Columbia Rivers in the United States, each hosting over 50 non-native fish species. The three most widespread non-native species globally are *Cyprinus carpio*, *Oncorhynchus mykiss*, and *Gambusia affinis*, of all which have established in over 50 countries and more than 200 river basins. Across different biogeographical realms, exotic fishes have colonized more than half of the river basins in Indo-Malay (64.71%), Palearctic (57.89%), and Australasia (53.52%). In the Nearctic, Neotropic, Oceania, and Afrotropic realms, the percentage of colonized river basins exceeds 30%. In contrast, translocated fishes have colonized 14.12%, 21.20%, 12.83%, 51.01%, 13.38%, 0.00% and 12.41% of river basins in these seven biogeographical realms, respectively (Fig. 1). Specifically, the southern and central Nearctic, northern and southern Neotropic, western and southern Palearctic, southern Afrotropic, northern Indo-Malay, and southern Australasia regions host a greater number of exotic fish species compared to other parts of the world (Fig. 2a, b). In contrast, the Nearctic and central Palearctic regions have more translocated fish species (Fig. 2e, f). In terms of individual countries, the United States leads the list with 302 non-native fish species, followed by Canada (63), Brazil (60), Russia (58), Mexico (56) and China (53) (Fig. S2a, b). Exotic fish species represent 8.48%, 10.90%, 0.88%, 9.70%, 3.49% and 1.25% of all fish species in these countries, respectively, while translocated fish species account for 22.00%, 14.60%, 0.77%, 11.90%, 3.25% and 4.28%, respectively (Fig. S2c, d).

### Relationship between phylogenetic relatedness and occurrence probability

Across the globe, the occurrence likelihood of non-native fish species significantly decreased when these species were less closely related to the native fishes (Fig. 3a–d). This trend held true whether these non-native species were introduced from foreign countries or translocated from different river basins within the same country, regardless of whether mean or nearest phylogenetic distances were employed to assess the phylogenetic relatedness. These relationships remained robust even after statistically accounting for phylogenetic independence (Table S1) and the potential influence of basin area (Table S2). This finding was further confirmed after excluding the non-native species within the two dominant families, Cyprinidae and Salmonidae (Fig. S3), and after redefining the exotic and translocated non-native species on the biogeographical realm scale (Fig. S4). The negative relationships between the occurrence probability and phylogenetic distance held true for both exotic and translocated fish species in each biogeographical realm (Fig. 4), and in the six countries with the largest number of non-native fishes (Fig. S5).

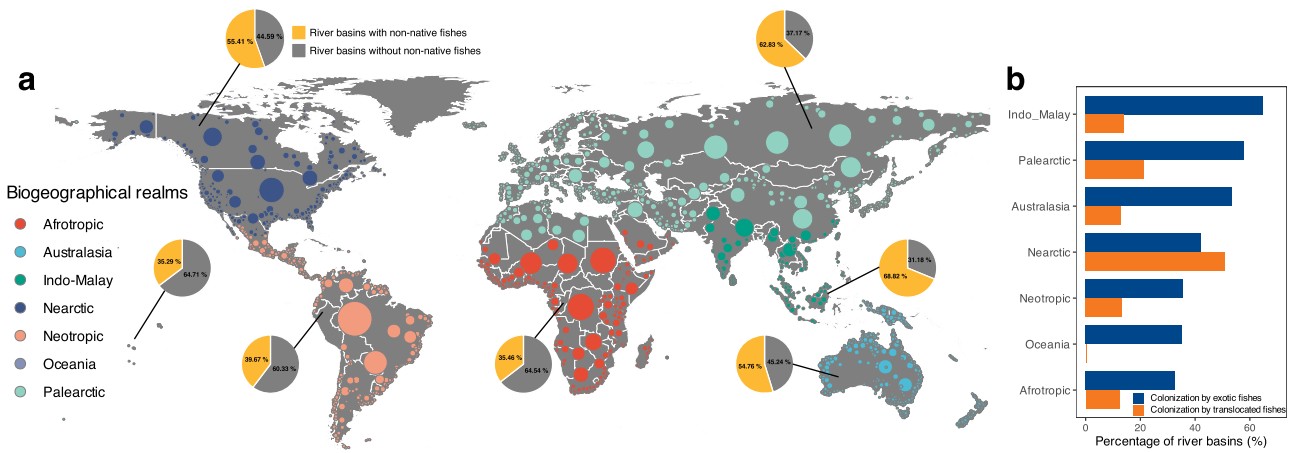

**Fig. 1 | Sampling river basins and non-native fish colonization patterns.**
**a** Geographical distribution of 3119 river basins across seven biogeographical realms and percentage of river basins colonized by non-native fish species in each realm. **b** Percentage of river basins colonized by exotic and translocated non-native fish species in each realm. The locations of river basins are represented by the median points with point size indicating basin area.

## The direct and indirect effect of phylogenetic relatedness and native diversity on the occurrence probability

The results of structural equation modeling demonstrated that the negative direct effects of phylogenetic distance on both exotic and translocated fish species remained robust and consistent even after accounting for the influences of native species richness and phylogenetic diversity (Figs. 5a, b and S6). Native species richness consistently exhibited positive effects on non-native fish occurrence, indicating that as the native species richness increased, there was a corresponding and consistent increase in the probability of non-native species occurrence (Figs. 5a, b and S6 and Table S3). Moreover, it is worth noting that higher levels of species richness appeared to be indirectly linked to the non-native species occurrence by predicting a reduction in the phylogenetic distance between non-native and native species (Fig. S6). The direct effects of native phylogenetic diversity on the occurrence of non-native fishes varied depending on the diversity indexes employed. The mean distance index (MPD) showed a positive effect (Fig. 5a, b and Table S3), indicating that higher MPD values are associated with an increased likelihood of non-native fish occurrences. In contrast, the nearest index (MNTD) displayed a negative effect (Fig. S6 and Table S3), suggesting that greater MNTD values are linked to a decreased likelihood of non-native fish occurrences. However, irrespective of the diversity index employed, native phylogenetic diversity consistently had negative indirect effects on non-native fish occurrences, primarily mediated through the nonnative-native phylogenetic relatedness (Figs. 5a, b and S6).

## Discussion

Drawing on an updated global biogeography of exotic and translocated non-native fish species, our study revealed a notably elevated occurrence probability of non-native fish species when they were closely related to native species, irrespective of at global, biogeographical realm, or country scales. Our findings suggest that native fish communities hosting close relatives may be particularly favorable for non-native fishes. The habitat adaptation advantages they offer outweigh the potential negative impacts from intensified competition, ultimately promoting the establishment of non-native fish species. The global patterns observed in non-native freshwater fishes and their phylogenetic associations could establish a basis for comprehending and forecasting future fish invasions in freshwater ecosystems.

We showed that 601 non-native freshwater fish species have successfully colonized over half of the river basins globally. Among these, exotic fishes have established in nearly 50% of these river basins spanning all the biogeographical realms. In contrast, translocated

fishes have colonized ~20% of these river basins, with a notable concentration in the Nearctic and Palearctic realms of the northern hemisphere. Given that the ongoing introductions due to the growing aquaculture and ornamental trade[6,34], it is evident that most rivers worldwide face serious threats from non-native fish invasions. This necessitates urgent assessments of their ecological and economic impacts on freshwater ecosystem and the implementation of effective control measures. It's worth noting that the number of non-native fish species identified in our study exceeds the recent record of 551 non-natives[6]. This discrepancy likely arises the common omission of translocated fishes within a country by previous studies[13]. Given that such intra-country translocations have occurred in over 600 river basins worldwide, our research emphasizes the pressing need to incorporate them into distribution predictions of non-native fishes and evaluate their potential influences on resident fish species and ecosystems.

Our analysis revealed a consistent and widespread negative relationship between nonnative-native phylogenetic distance and the occurrence of non-native fish species. In other words, non-native fishes were more likely to establish themselves in river basins where native species closely related to them were already present. This pattern held true for both exotic and translocated species across global, biogeographical, and country scales, regardless of the phylogenetic relatedness index adopted. Importantly, this relationship remained robust even after accounting for the influence of native species richness, native phylogenetic diversity, and basin area. Our results therefore substantially extend the previous finding that a close phylogenetic relatedness predicts non-native fish success in regional freshwater lakes[31], establishing a general and global pre-adaptation pattern for freshwater fish species. However, our results contrast with an earlier observation that found no evidence supporting the pre-adaptation hypothesis in explaining fish invasions[24]. The discrepancy may be attributed to differences in how relatedness was measured, and the spatial-temporal scales considered. For example, Ricciardi and Mottiar used the presence of congeneric native species to represent close relatedness rather than quantifying specific phylogenetic distances[24]. The method based on taxonomic classification assumes that relatedness would remain constant between genera and all congeneric species are equally related, potentially limiting the precision of their conclusions[21]. Additionally, while Ricciardi and Mottiar examined congeneric species at the country and regional scales[24], we measured phylogenetic distance for specific lakes or rivers, and the differences in spatial scale may partly explain this discrepancy[27]. Collectively, our findings strongly support the pre-adaptation hypothesis, indicating

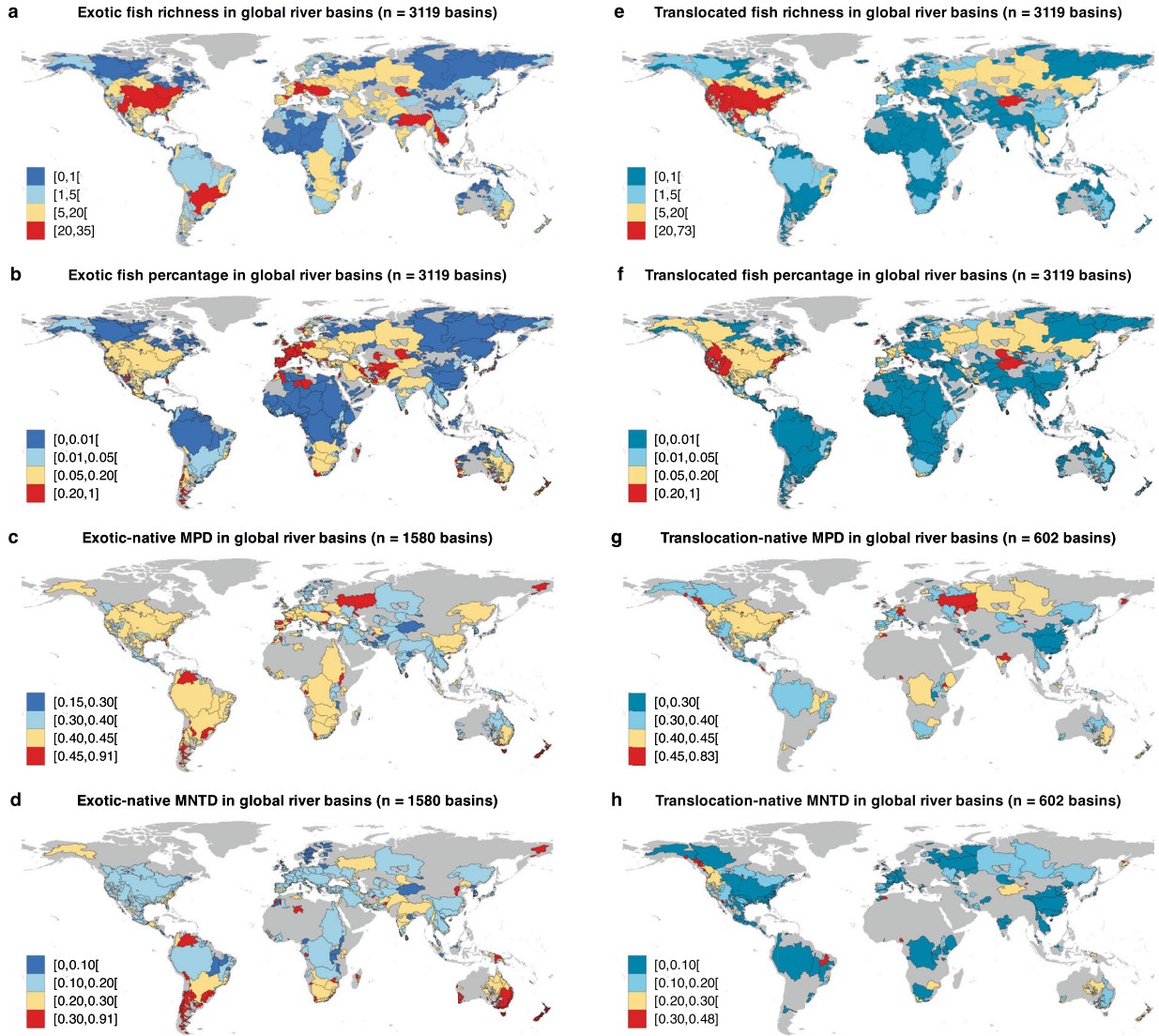

**Fig. 2 | Global geographical distribution of exotic and translocated freshwater fish species. a** Geographical pattern of exotic fish richness. **b** Geographical pattern of exotic fish percentage. **c** Mean phylogenetic distance (MPD) between exotic and native fish species in river basins where exotic fish species occur. **d** Nearest phylogenetic distance (MNTD) between exotic and native fish species in river basins where exotic fish species occur. **e** Geographical pattern of translocated fish richness. **f** Geographical pattern of translocated fish percentage. **g** Mean phylogenetic distance (MPD) between translocated and native fish species in river basins where translocated fish species occur. **h** Nearest phylogenetic distance (MNTD) between translocated and native fish species in river basins where translocated fish species occur. The percentage represents the ratio of non-native species richness to the total species richness in each river basin. The number of river basins used for assessing these patterns is displayed at the top of each panel.

that the presence of close relatives predicts the occurrence of non-native fish species in freshwater ecosystems. These patterns contribute to a growing body of evidence suggesting that phylogenetically close non-native animal species are more likely to establish successfully[29,35,36], emphasizing that adaptation to abiotic environments might play a more pivotal role than competition in explaining the establishment of non-native animal species.

One possible explanation for the positive association between close phylogenetic relatedness and the occurrence of non-native fish species, in contrast to the equivocal findings in plant species[26,27,37,38], may be attributed to the mobility and cognitive abilities of non-native fishes. These attributes allow them to select habitats with lower resource competition and thus their ability to adapt to novel environmental conditions in new regions may be a critical factor in determining their success or failure. Supporting this, Moyle and Light found that abiotic factors, rather than competition, primarily determined the establishment of non-native fishes in California streams[15].

Furthermore, considering that phylogenetic relatedness may capture similarities in fish physiological traits such as thermal tolerance[39] and hypoxia tolerance[40], closely related non-native fishes may adapt better to new habitats due to their physiological similarity to native species. Consequently, they tend to be more successful. However, it should be noted that our study primarily focused on large spatial scales, which may have a substantial influence on our findings. At local spatial scales, it is commonly hypothesized that biotic interactions play a more critical role, which would align with predictions from Darwin's naturalization hypothesis. Conversely, at regional scales, such as those considered in this study, environmental filtering is believed to be more important, leading to results consistent with predictions of the pre-adaptation hypothesis[21]. Owing to our data structure, we were unable to explicitly distinguish the effects of region scales from those of local scales in this study. Nevertheless, our results unequivocally showed that in the natural context of river basins where native and non-native fishes coexist and interact, the presence of close relatives predicts

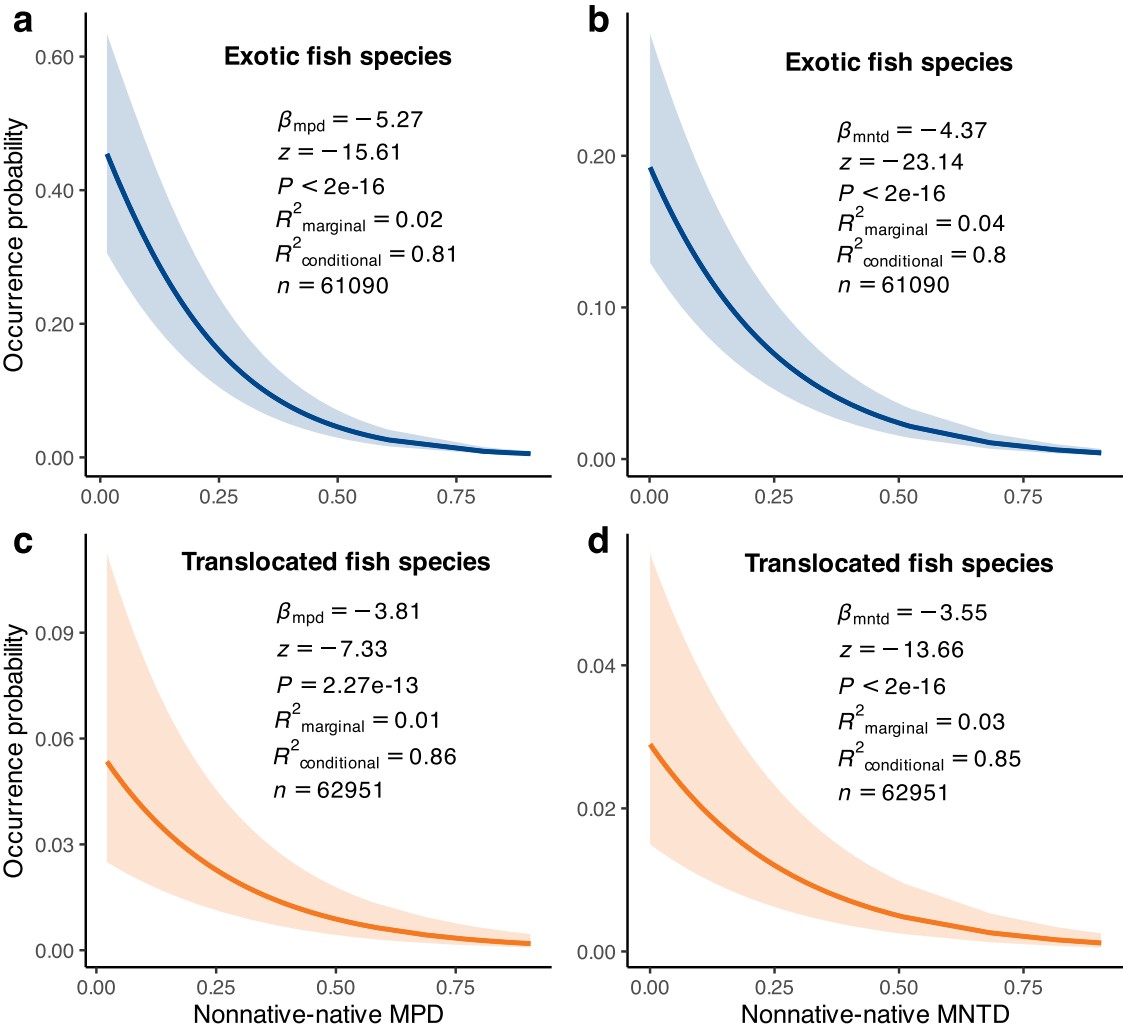

**Fig. 3 | Relationships between the probability of non-native fish occurrence and nonnative-native phylogenetic distance across 3008 river basins worldwide.** **a** The variation in the occurrence probability of exotic fish species with the mean phylogenetic distance (MPD) between exotic and native fish species. **b** The variation in the occurrence probability of exotic fish species with the nearest phylogenetic distance (MNTD) between exotic and native fish species. **c** The variation in the occurrence probability of translocated fish species with the MPD between translocated and native fish species. **d** The variation in the occurrence probability of translocated fish species with the MNTD between translocated and native fish species. Statistical tests and predictive curves (with 95% confidence intervals) were obtained using generalized linear mixed models (GLMMs), while assuming a binomial error distribution. Statistical significance (p values), variance explained ($R^2_{marginal}$ for the fixed effect and $R^2_{conditional}$ for both the fixed and random effects), and sample size (n) are presented in the figure. Blue and orange colors are used to highlight the relationships observed in exotic and translocated fish species, respectively.

higher success of non-native fishes. Our findings indicate that phylogenetic relatedness between non-native and native fishes can provide valuable insights into understanding and predicting fish invasions in freshwater ecosystems. We urge future studies to comprehensively assess the influence of phylogenetic relatedness on fish invasions across diverse spatial scales and environmental gradients and to compare the findings with results from other taxonomic groups, which will contribute significantly to a comprehensive understanding of the roles of phylogenetic relationship in predicting biological invasions.

We also observed consistent positive associations between native species richness and the occurrence of both exotic and translocated fish species. This implies that a higher native fish richness does not hinder their invasion but rather predicts a greater likelihood of non-native fish occurrence. Moreover, we found that high native richness indirectly associated non-native fish establishment by influencing nonnative-native phylogenetic relatedness. These findings do not support the biotic resistance hypothesis, which suggests that non-native species would have lower success in highly diverse communities[32,41]. Instead, they align with the predictions of the biotic

acceptance hypothesis[42,43]. The positive species diversity-occurrence relationship, consistent with the relatedness-occurrence relationship described in earlier sections, suggests that competition between non-native and native fish species probably plays a minor role in determining the establishment of non-native fish species. Rather, high species richness and closely related native species within native communities may represent a suitable environment favoring non-native fishes as well as the natives, thereby predicting their success. However, it is worth noting that the positive species diversity-occurrence relationship may also be attributed to the isolation and size of the river basin, as predicted by the theory of island biogeography[44]. While more isolated and small river basins tend to have lower native diversity, they may also support fewer non-native species. This aspect warrants further investigation in future studies. In contrast to species diversity, the effect of native phylogenetic diversity was more intricate and highly dependent on the chosen phylogenetic indices. While high phylogenetic diversity based on the nearest distance predicted a lower probability of non-native fish occurrence, the pattern was reversed when mean phylogenetic distance was used. These results indicate that

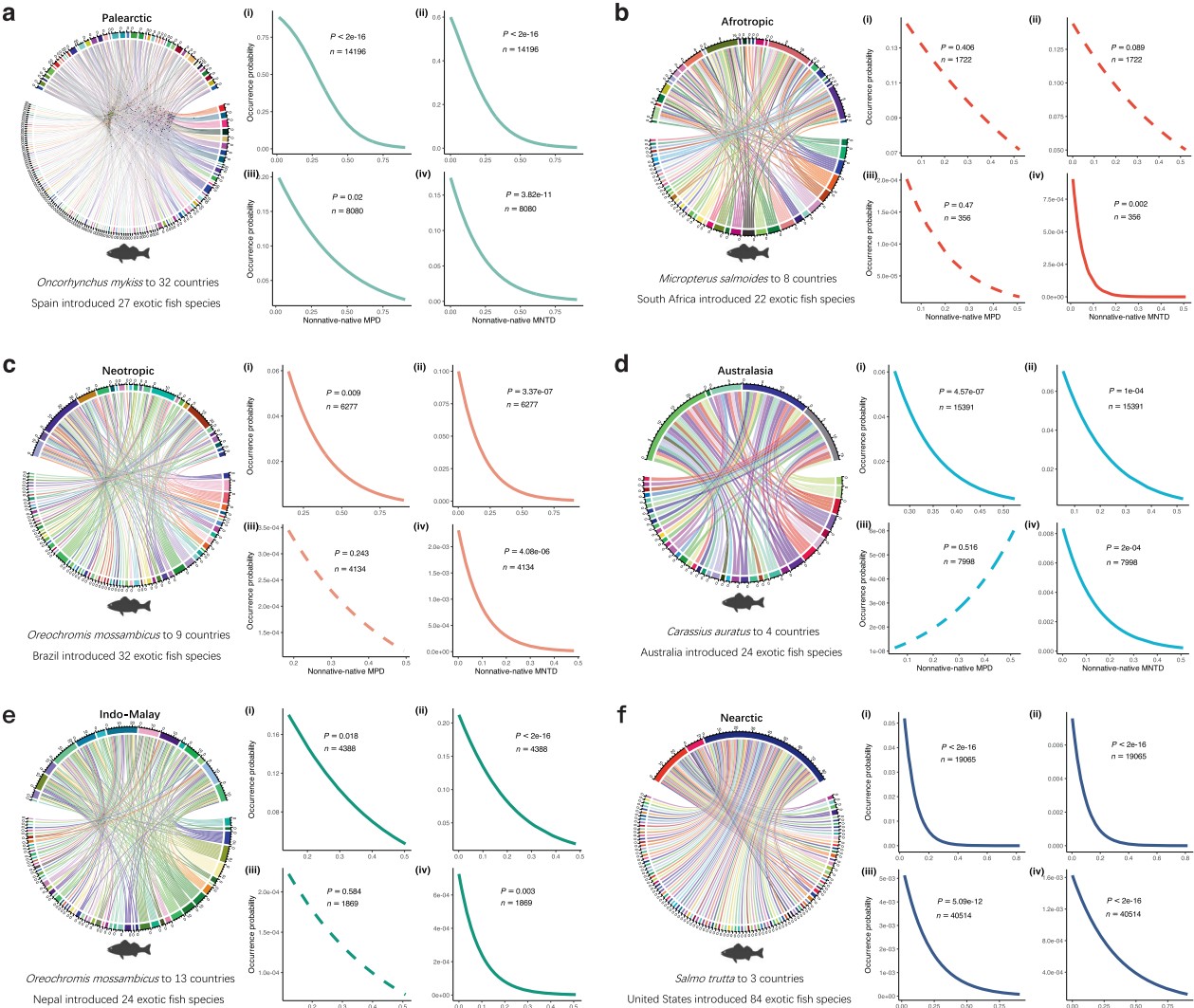

**Fig. 4 | Relationships between the probability of non-native fish occurrence and nonnative-native phylogenetic distance in six biogeographical realms of the world. a** The relationship in the Palearctic realm. **b** The relationship in the Afrotropic realm. **c** The relationship in the Neotropic realm. **d** The relationship in the Australasia realm. **e** The relationship in the Indo-Malay realm. **f** The relationship in the Nearctic realm. For each biogeographical realm, (i) the variation in the occurrence probability of exotic fish species with the MPD between exotic and native fish species, (ii) the variation in the occurrence probability of exotic fish species with the MNTD between exotic and native fish species, (iii) the variation in the occurrence probability of translocated fish species with the MPD between translocated and native fish species, and (iv) the variation in the occurrence probability of translocated fish species with the MNTD between translocated and

native fish species. Statistical tests and predictive curves were obtained using generalized linear mixed models (GLMMs) while assuming a binomial error distribution, with solid lines representing significant effects ($p < 0.05$). Statistical significance ($p$ values) and sample size ($n$) are displayed in the figure. Different colors are used to highlight the relationships observed in different biogeographical realms. For each realm, a chord diagram illustrates the network relationship between countries and exotic fish species, showing which exotic fishes have been introduced into specific countries and which countries have introduced specific exotic fish species. The most frequently introduced exotic fish species and the country that introduced the largest number of exotic fish species are noted below the chord diagrams for clarity.

understanding biotic resistance in fish communities may be more complicated than initially thought, emphasizing the importance of considering different dimensions of diversity when assessing the relationship between biodiversity and biological invasions[1,33]. Nevertheless, it is clear that these mixed roles of native diversity do not alter the relationship between phylogenetic relatedness and non-native fish occurrence. Non-native fishes that are phylogenetically close to native species consistently exhibit a higher likelihood of establishment, regardless of the taxonomic and phylogenetic diversity present in the river basins.

Several limitations in our study should be acknowledged. Firstly, identifying translocated non-native species from native ones remains a formidable challenge. Determining definitively whether a species has been translocated among river basins or has a historical presence in a

particular basin is often impossible, primarily due to the extensive connectivity among basins and the scarcity of historical records. In this study, direct observations of translocated fish species were not conducted. Instead, we classified a species as translocated when it was identified as non-native in a specific river basin while simultaneously being recorded as a native species in other river basins within the same country. The absence of clear records for translocated species may introduce bias into the geographical patterns we identified, potentially influencing our analysis of the underlying drivers. For instance, our findings indicated a predominant occurrence of fish translocations in Nearctic and Palearctic realms of the northern hemisphere. However, it is plausible that the majority of the sampling effort was concentrated in these regions, implying a potential bias in our current results. While an increasing number of studies recognized the importance of

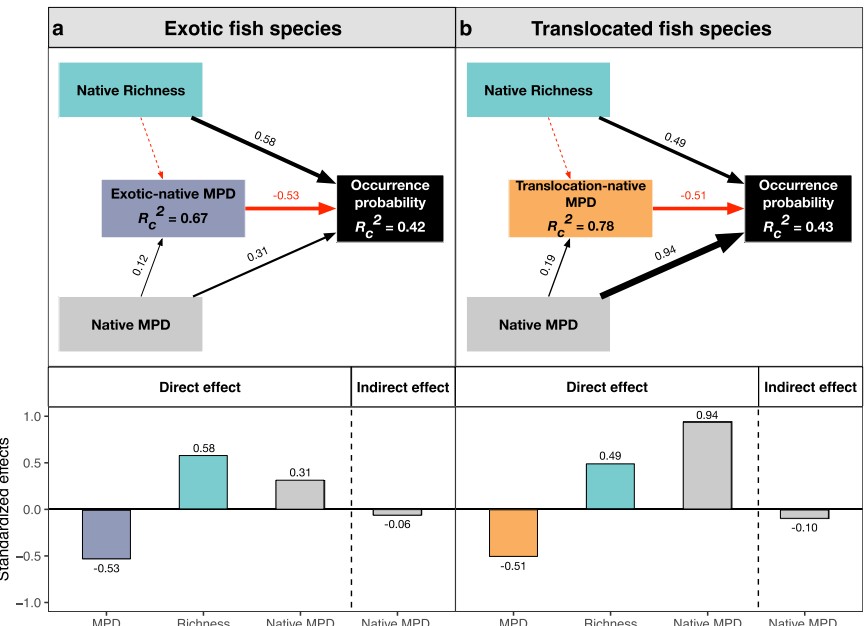

**Fig. 5 | Bayesian structural equation modeling for assessing direct and indirect effects of phylogenetic relatedness, native species richness, and native phylogenetic diversity on the probability of non-native fish occurrence. a** The relationship among exotic-native MPD, native richness, native MPD, and the occurrence probability of exotic fish species. **b** The relationship among translocation-native MPD, native richness, native MPD, and the occurrence probability of translocated fish species. Boxes represent measured variables, highlighted with distinct colors, while arrows represent relationships among variables. Black and red arrows denote positive and negative effects, respectively. Dashed and solid lines denote 95% credible intervals overlapping with zero or not, respectively. Standardized path coefficients are provided for each significant path, with the width of the path scaled to reflect the magnitude of the standardized path coefficient. The Bayesian conditional $R_c^2$ (based upon both fixed and random effects) for the endogenous variable is reported in the corresponding boxes. The direct and indirect effects are calculated and presented in the lower part of each panel.

identifying translocated species[1,8,13,14], it is crucial for future research to prioritize the explicit and extensive documentations of species translocation among different river basins. Secondly, the introduction preferences of non-native fishes may obscure the associations between phylogenetic relatedness and non-native fish occurrence that we have identified. On the one hand, non-native fishes are often intentionally introduced for purposes like aquaculture and recreation[6], and certain species may be introduced more frequently with larger numbers of individuals (i.e., high propagule pressure)[45]. This could result in more opportunities for them to become established, potentially influencing our results related to competition and environmental filtering. On the other hand, the absence of non-native fishes in a river basin could be due to either a lack of introduction events or failure to establish. Failing to differentiate between these possibilities may lead to an overestimation of the number of establishment failures and influence the precision of our conclusions[24]. Ideally, these issues could be addressed by conducting studies across different invasion stages[26,46] or by focusing on a specific stage with clear establishment failure data[28,31]. Due to our data structure, we were unable to distinguish between these stages in this study. We strongly advocate for future studies in freshwater ecosystems that can explicitly differentiate and evaluate the effects of phylogenetic relatedness during the introduction, establishment, and impact stages. Finally, it's worth noting that our comprehensive and intricate database may exhibit uneven data distributions and potential extreme outliers, which could have also influenced our results.

In summary, through our comprehensive and up-to-date exploration of the biogeographical patterns of non-native freshwater fish species across the globe, we unveiled the pervasive influence of phylogenetic relatedness in shaping their distribution. These findings lay a foundation for evaluating the worldwide impacts of non-native fishes and predicting potential fish invaders in the future. The global hotspots identified in this study should be considered high-priority areas for assessing ecological threats, economic costs and benefits associated with non-native fishes[47,48]. They should also inform policymakers and managers in developing public policies addressing freshwater fish invasions. We demonstrated that the presence of close relatives and high species richness, rather than impeding invasion, are indicative of a higher likelihood of non-native fish occurrence. These results suggest that, compared to competition with native fish species, the ability to adapt to novel environments may play a more pivotal role in determining freshwater fish invasions. A promising direction for future research is to delve deeper into assessing the relative importance of phylogenetic relatedness in comparison to other potential factors, such as human activities and functional characteristics, and to examine the variation in the effect of phylogenetic relatedness across multiple spatial scales and different invasion stages. These further investigations will substantially contribute to a deeper and more comprehensive understanding of global freshwater fish invasions in the Anthropocene.

## Methods

### Occurrence data for global freshwater fish species
We employed a highly comprehensive global database of freshwater fish species distributions, meticulously documenting the occurrence of 14,953 species in 3119 river basins, covering more than 80% of the Earth's continental surface[12]. This database has been instrumental in exploring the taxonomic, functional, and phylogenetic diversity of freshwater fishes on a global scale and advancing our understanding of human impacts on freshwater fish diversity[1,11,49]. In this database, each river basin was assigned to one of the 143 countries (or the primary country for shared river basins), which are further nested within seven biogeographical realms (i.e., Afrotropical, Australasia, Indo-Malay, Nearctic, Neotropic, Oceania, and Palearctic)[50]. For each basin, geographic coordinates of its centroid and the surface area were also provided, enabling us to locate all these river basins across the seven

biogeographic realms (Fig. 1a). The database records the occurrence of fish species in each river basin, representing all the freshwater fish species inhabiting the entire river network of that basin. Each fish species in each river basin is categorized as either native or non-native, with a non-native species defined as an introduced species that has completed its life cycle and established self-sustaining populations within that basin[12].

## Distinguishing exotic and translocated fish species

We categorized non-native fish species into two groups based on their geographical origins, in accordance with previous studies[13,14,51]: exotic species originating from other countries and translocated species within the same country. We identified translocated fish species by checking whether a non-native species in a particular river basin within a country had also been observed as a native species in other basins within the same country. When such instances were found, it was assumed that the non-native species had been translocated from those other basins (it should be noted, however, that occasionally these non-natives may still have been directly introduced from another country). Exotic fish species were determined by excluding translocated species from the non-native species lists of each river basin. For each biogeographical realm, we examined the percentage of river basins that had been colonized by exotic fish species and those that had been colonized by translocated non-native fishes. Additionally, we assessed the richness and the percentage of both exotic and translocated non-native fishes in each river basin, aiming to represent their occurrence and distribution across the world.

To more accurately depict the natural distribution of non-native fish species while minimizing the influence of administrative boundaries, we also broadened our analysis from a country-level perspective to a biogeographical realm scale. This allowed us to redefine the categories of exotic and translocated fish species, in accordance with previous studies[1,8]. Specifically, we defined exotic species as those originating from different biogeographical realms, and translocated species as those moved within the same realm. We then compared the relationships between the occurrence likelihood of non-native species and nonnative-native phylogenetic relatedness under these two distinct definitions.

## Phylogenetic relationship for global freshwater fish species

We constructed a global phylogenetic tree for freshwater fish species using the *FishPhyloMaker* R package[52]. This package creates a phylogenetic tree for fish species by incorporating and pruning species from a backbone phylogenetic tree[53]. Initially, we employed the *FishTaxaMaker* function to generate 14,892 valid species names, excluding 61 duplicate names from the initial list of 14,953 species. These valid names were then used as input for the *FishPhyloMaker* function, resulting in a phylogenetic tree that encompassed 14,708 fish species, with 184 species names automatically excluded as unidentifiable. This final phylogenetic tree of 14,708 species represents the evolutionary relationships among freshwater fish species worldwide, including 597 species introduced as either exotic or translocated species. Using this phylogeny, we calculated pairwise phylogenetic distances among all species using the *cophenetic* function of the *ape* R package[54].

## Nonnative-native phylogenetic relatedness and native diversity metrics

Based on the computed pairwise distances, we calculated two widely used phylogenetic distance metrics to represent the phylogenetic relatedness between non-native and native species[27,29]: (1) The nonnative-native mean phylogenetic distance (MPD), which measures the average distance between a non-native fish species and all native fish species within a specific river basin. With this measure, we assume that each native fish species within a river basin equally contributes to the occurrence of non-native fish species. (2) The nonnative-native

nearest taxon distance (MNTD), which quantifies the phylogenetic distance between a non-native fish species and its closest native relative in a river basin. This metric, in contrast, assumes that the presence or absence of a non-native fish species in a river basin is primarily influenced by its proximity to its closest native relative, as they are more likely to utilize similar resources and share common enemies and mutualists[26]. For each river basin, we further assessed the phylogenetic diversity of native species by employing two additional metrics with the *picante* R package[55]: native MPD (calculated by mean pairwise phylogenetic distances among all the native species in a river) and native MNTD (computed by mean phylogenetic distance of each native species to its nearest native neighbor in a basin)[56,57]. In addition, we calculated native species richness to characterize the overall species diversity within each river basin. These combined approaches allowed us to examine the relationships between nonnative-native phylogenetic relatedness and whether non-native fishes were present or absent, while considering the influences of both the phylogenetic and species diversity of the native communities.

## Nonnative-native phylogenetic relatedness and occurrence likelihood of non-native fishes

To assess the impact of nonnative-native phylogenetic relatedness on the occurrence of non-native fish species in a river basin, we refined the occurrence data according to the following criteria: (1) We excluded 22 countries where no non-native species were recorded. We required that each country had at least one established non-native fish species in its river basins. This criterion was essential because the absence of any occurrence records would make it impossible to evaluate the phylogenetic distance between non-native and native fishes. (2) We excluded species that were not represented in the constructed phylogenetic tree, including six translocated non-native species. These species were omitted because their absence in the phylogenetic tree prevented the characterization of their evolutionary relationships. Consequently, we used a cleaned occurrence dataset, which included 14,708 species across 3008 river basins in 121 countries, to analyze the relationship between phylogenetic relatedness and the occurrence of non-native fish species.

For each exotic fish species occurring in a specific country, we categorized its presence in a river basin as a success (assigned a value of 1) and its absence as a failure (assigned a value of 0). We then calculated its MPD and MNTD with all native fish species in each river basin within the country, regardless of whether the exotic fish was present in that specific basin. Similarly, for each translocated fish species within a country, we considered its presence in a river basin as a success (1) and its absence as a failure (0). We also calculated its MPD and MNTD with all native fish species in each river basin, except for the basin from which the translocated fish originated (Fig. S7). This approach allowed us to establish the connection between exotic-native phylogenetic distances and whether the exotic fishes occur (total $n = 61,090$ records), as well as translocation-native phylogenetic distances and whether the translocated fishes occur (total $n = 62,951$ records). Subsequently, we examined the relationship between phylogenetic relatedness and the probability of non-native fish occurrence.

## Statistical analysis

We modeled occurrence probability (presence or absence of non-native species) as a function of phylogenetic distance using generalized linear mixed models (GLMMs) assuming a binomial error distribution and logit link function. In cases where the binomial response variable had a significantly higher number of zeros than ones, we opted for the clog-log link function, as recommended[58]. It's worth noting that the choice of link function did not substantially alter the results and was primarily employed to mitigate potential bias. To test the effect of phylogenetic relatedness on a global scale and within different

biogeographical realms, we included MPD and MNTD as separate fixed predictors, with species (1|species) and river basin nested in the country (1|country/basin) treated as random effects. These random effects were used to account for the statistical non-independence of multiple presence/absence records of a specific non-native species and the multiple records in a specific river basin within a country. To examine the effect of phylogenetic relatedness within a specific country, similar GLMMs were used with species (1|species) and river basin (1|basin) treated as random effects. These GLMMs were separately applied to exotic and translocated species and were executed using *the glmer* function of the *lme4* R package[59]. We also computed the explained variance by the fixed effect ($R^2_{marginal}$) and both the fixed and random effects ($R^2_{conditional}$) using *the r2* function of the *performance* R package[60], which follows the methodology developed by Nakagawa and Schielzeth[61].

Considering that Cyprinidae and Salmonidae include the most common non-native fish species, and with the former showing a preference for establishing populations in warm rivers while the latter primarily thrive in cold rivers, the environmental preferences of these key non-native species may have a substantial impact on our current findings, potentially limiting their generalizability. To address this concern, we conducted additional analyses by excluding data related to non-native species within these two dominant families. Subsequently, we re-evaluated the relationship between the occurrence probability and phylogenetic relatedness across the globe.

Following the redefinition of exotic and translocated non-native species on the biogeographical realm scale, we obtained the corresponding relationship between exotic-native phylogenetic distances and whether the exotic fishes occur (total $n = 126,419$ records), as well as translocation-native phylogenetic distances and whether the translocated fishes occur (total $n = 281,649$ records) and re-examined the impact of phylogenetic relatedness on the occurrence likelihood of non-native fish species at the global scale.

To account for the phylogenetic non-independence among non-native fish species, we also conducted Bayesian phylogenetic mixed models. These models encompassed the same fixed and random effects as the aforementioned GLMMs, while incorporating an additional phylogenetic covariance structure into the models. We constructed the phylogenetic covariance matrix for the global fish phylogeny using the *vcv* function of the *ape* R package[54] and extracted the sub-matrices specific to exotic and translocated species, respectively. These models were fitted using the *INLA* R package with default priors[62], which uses integrated nested Laplace approximation for Bayesian inference. This method allows for the rapid approximation of Bayesian posterior distributions and accommodates complex layered random effects, including autocorrelation terms[63]. While the results of the Bayesian phylogenetic mixed model were very similar to those from GLMMs, we presented them in the supplementary materials (Table S1).

We also conducted Bayesian structural equation modeling (BSEM) using a Bayesian multivariate response model with the *bf* function of the *brms* R package[64]. This analysis aimed to examine the robustness of the effect of phylogenetic relatedness while considering the influences of native diversity, evaluate the support for biotic resistance hypotheses, and explore their direct and indirect effects on non-native fish occurrence. BSEM can integrate a set of structural equations (component models) including random effects, allow for non-Gaussian error distributions, and assess multiple relationships simultaneously, making it suitable for our data structure. Specifically, we incorporated species (1|species) and river basin nested in the country (1|country/basin) as random effects in each component model to account for sampling non-independence. For the endogenous response variables, which were the presence/absence of non-native fish species and nonnative-native phylogenetic relatedness, we hypothesized Bernoulli and Gaussian distributions, respectively. Prior to including them in the BSEM model, we standardized all continuous phylogenetic relatedness and diversity metrics using the $z$-score (by subtracting the mean and dividing by the standard deviation). This allowed us to obtain standardized parameter estimates (path coefficients), calculate direct and indirect effects, and interpret them on a comparable scale. We verified that the inclusion of both relatedness and diversity metrics did not lead to multicollinearity issues because there were low correlations among them (Fig. S8) and *VIFs* (variance inflation factors) for all metrics remained below three. The mean and credible interval of parameters were estimated based on posterior values derived from 4 chains of 2000 iterations, with the first 1000 steps as burn-in. The default weakly informative priors were used for all the fixed and random effect parameters. We confirmed a good chain convergence according to *R-hat* (the potential scale-reduction factor), with values consistently below 1.01 for all models[64].

Lastly, the likelihood of non-native fish occurrence may be highly correlated with the size of the river basin, potentially confounding the results described above. Therefore, we conducted additional checks on the relatedness-occurrence and diversity-occurrence relationships by incorporating basin area as a covariate in the GLMMs. All the statistical analyses were performed in R v4.3.0[65].

### Reporting summary

Further information on research design is available in the Nature Portfolio Reporting Summary linked to this article.

## Data availability

The data that support the findings of this study are available at https://doi.org/10.6084/m9.figshare.25058651. Original freshwater fish occurrence database is available at https://doi.org/10.6084/m9.figshare.c.3739145.v1.

## Code availability

R codes that support the findings of this study are available at Figshare[66] with the identifier https://doi.org/10.6084/m9.figshare.25058651.

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

## Acknowledgements

This work was supported by Central Public-interest Scientific Institution Basal Research Fund, CAFS (2023TD17) to M.X., Guangdong Provincial Special Fund for Modern Agriculture Industry Technology Innovation Team (2022KJ134) to M.X., the China Agriculture Research System of MOF and MARA (CARS-45) to D.G., the Science and Technology Program of Guangzhou, China (2023B03J1306) to M.X., the National Freshwater Genetic Resource Center (FGRC18537) to X.M.

## Author contributions

M.X. conceived the study design, conducted the analyses, and wrote the first draft of the manuscript. S.L., C.L., P.T. and J.D. interpreted the results and edited the following versions. D.G., X.M., M.F., H.W., F.Y., L.S. and X.W. contributed substantially to revisions.

## Competing interests

The authors declare no competing interests.
