## [Peer Review File · Nature Communications]

Global freshwater fish invasion linked to the presence of closely related speciesReviewer #1 (Remarks to the Author):

This manuscript, based on a novel reanalysis of an existing global fish dataset, is an important contribution to the invasiveness-invasibility discussion.

The key finding is that "non-native fishes were more likely to occur in river basins where native close relatives are present". However, many widespread invasive species are from a limited number of families and those families have a lot of species, potentially biasing this result or at the very least the level of confidence in "determining" (ln. 489) vs. "being associated". Perhaps this merits additional discussion or even analyses.

It is essential to present raw data graphically, e.g. as univariate relationship figures with data points. When all data are derived, it is difficult to assess the relative effect of an under-sampled part of a particular gradient, for example.

The problem with intra-basin translocations is that it is often impossible to tell whether a species has truly been translocated anew or has been historically present in that part of the basin. Only a few basins have long-term geological features blocking migrations. Intra-country translocations seem to be even more problematic to define, as they could be a mix of true invasions and previously connected basins. Perhaps this could be discussed a bit more.

The discussion of translocation hotspots seems to come out of nowhere (not prominent in the results or study set-up).

Another recent study (Qian et al. 2022), demonstrated that "Support for Darwin's naturalization and preadaptation hypotheses in freshwater fish assemblages depends on the phylogenetic diversity of assemblages within the invaded watersheds." It may be relevant to cite this work and possibly additional references mentioned therein in this context.

Finally, while the original data used in this study have already been published, it is important for authors to fully release derived data and indices.

Minor comments:

In 57 what is "their" referring to?

In 68-9 sentence needs work

In 75-6 sentence needs work

In 77 which traits?

In 348-9 Unexpected opening for the discussion not focusing on the results of this study, especially since it was done on the existing data

Fig 2 While maps are insightful, it would be good to show univariate relationships for all of the variables of interest as well (e.g., alien richness vs. alien-native MPD, alien % vs. alien-native MPD; same for co-variates). At present it is not possible to tell how much these results (including figure 3) are affected by unequal sample distribution across gradients

Fig 5 is difficult to interpret

Fig 6(a) the effect of native fish richness is as strong or stronger than the effect of MPD, why is the arrow thickness not aligned with that effect size? Why does the model not account for covariation between native richness and native MPD?

Reviewer #2 (Remarks to the Author):

The manuscript by Xu et al. investigates the influence of phylogenetic relatedness on invasion success of non-native fish in more than 3000 drainages across the world. Additionally, the study investigates differences in alien fish, which originates from outside- and inside a given country (latter defined as translocated species). Lastly, the study also investigates the influence of local diversity on invasion success.

The overall theme definitely relevant, but how novel is it compared to previous publications? The Xu et al. 2022 GCB paper describes the same general finding, but for lakes (not rivers). The addition with a contrast between alien and translocated fish is an interesting addition, but has some caveats (see below).

Including fish species translocated between drainages within a country is definitely a strength of this study. However, it is a bit problematic that movement of fish species within and between countries are used as defining translocated- and alien fishes, respectively. Countries are made by political borders and translocations within countries may be between drainages with very different colonization history, whereas translocation between countries may be within the same major drainages. This is further problematic, since not all countries are of the same size, which will influence the use of the terms for different regions. This may very well explain the countries with the highest amount of translocated fish species are also among the largest in the world.

I can understand how MPD and MNTD is calculated for each non-native fish species and also for each of the native species present in a drainage. However, I cannot understand how this can also be used as a measure of diversity of native fish species (L193-195)?

I am also questioning how much of the results that are driven by some not do novel patterns, given that carp and rainbow trout are the most frequent alien species. It is not surprising that rainbow trout would mainly be able to establish populations in cold-water rivers, which are already the habitat of native salmonids, whereas carp would establish populations in warm, slow-flowing rivers, which are home to native cyprinids. It is then not really about ecological niche, but rather about general environmental requirements of these two important families. It would be good, if the authors could estimate how much of the effect that is caused by such e.g. cyprinid-salmonid environmental preferences.

Regarding the observed positive species diversity-occurrence relationship, I think that it is worth mentioning that it can also be explained by island biogeography (distance to the mainland) theory: if drainages are generally more isolated, they would tend to have more depauperate native diversity (e.g. colonization since last glaciation). Such more isolated drainages would potentially also tend to have less non-native species. But even drainage size could also influence this pattern: large drainages tend to have more native species, but also have more points of entry for alien species.

Response to reviewer's comments

Reviewer #1 (Remarks to the Author):

This manuscript, based on a novel reanalysis of an existing global fish dataset, is an important contribution to the invasiveness-invasibility discussion.

Response: Thank you for your positive evaluation of our manuscript and for your constructive comments, which have enhanced the quality of our manuscript.

Please find below our point-by-point response to your comments.

The key finding is that "non-native fishes were more likely to occur in river basins where native close relatives are present". However, many widespread invasive species are from a limited number of families and those families have a lot of species, potentially biasing this result or at the very least the level of confidence in "determining" (ln. 489) vs. "being associated".

Perhaps this merits additional discussion or even analyses.

Response: Indeed, many widely distributed non-native fish species belong to a limited number of families, leading to an uneven and independent sample distribution. The results obtained from the species of main families could potentially overshadow patterns in other species, thereby potentially limiting the generalizability of our findings.

To address this concern, we employed two approaches. On the one hand, we included species as a random effect in our statistical analysis and incorporated a phylogenetic covariance matrix to account for the phylogenetic correlations among these non-native species (Method: lines 492-495). This approach allows us to mitigate this issue statistically.

Lines 492-495: "To account for the phylogenetic non-independence among non-native fish species, we also conducted Bayesian phylogenetic mixed models. These models encompassed the same fixed and random effects as the aforementioned GLMMs, while incorporating an additional phylogenetic covariance structure into the models."

On the other hand, we conducted additional analyses to validate our results, as you suggested. Specifically, we excluded data involving non-native species from two main families, Cyprinidae and Salmonidae, which are known to comprise the most frequent non-native species. Remarkably, even after excluding these species, we still observed that non-native species were more likely to succeed when they were phylogenetically closely related to native fishes (Fig. S3), aligning with our previous analyses. This finding further strengthens our conviction that close relatedness to native species universally enhances the success of non-native species worldwide. We appreciate your valuable suggestion and have included these additional analyses into the Results section (lines 160-161) and Methods section (477-484).

Lines 160-161: “This finding was further confirmed after excluding the non-native species within the two dominant families, Cyprinidae and Salmonidae (Fig. S3)”.

Lines 477-484: “Considering that Cyprinidae and Salmonidae include the most common non-native fish species, To address this concern, we conducted additional analyses by excluding data related to non-native species within these two dominant families. Subsequently, we re-evaluated the relationship between the occurrence probability and phylogenetic relatedness across the globe.”

Fig. S3 | Relationships between the probability of non-native fish occurrence and nonnative-native phylogenetic distance after excluding the non-native species within the two dominant families, Cyprinidae and Salmonidae. a Occurrence probability of alien fish species with alien-native MPD. **b** Occurrence probability of alien fish species with alien-native MNTD. **c** Occurrence probability of translocated fish species with translocation-native MPD. **d** Occurrence probability of translocated fish species with translocation-native MNTD. Predictive curves (with 95% confidence intervals) were derived from generalized linear mixed models (GLMMs) assuming a binomial error distribution. Statistical significance (P values), variance explained (R^2_{marginal} for the fixed effect and $R^2_{\text{conditional}}$ for both the fixed and random effects), and sample size (n) are presented in the figure.

It is essential to present raw data graphically, e.g. as univariate relationship figures with data points. When all data are derived, it is difficult to assess the relative effect of an under-sampled part of a particular gradient, for example.

Response: We have supplemented the univariate relationship figures as suggested (Figure S8). Please also refer to our response to the following point (“Fig 2 While maps are insightful, it would be good to show univariate relationships for all of the variables of interest as well...”).

The problem with intra-basin translocations is that it is often impossible to tell whether a species has truly been translocated anew or has been historically present in that part of the basin. Only a few basins have long-term geological features blocking migrations. Intra-country translocations seem to be even more problematic to define, as they could be a mix of true invasions and previously connected basins. Perhaps this could be discussed a bit more.

Response: We acknowledge and appreciate your comment. Indeed, distinguishing translocated species from native species can be challenging due to the often limited historical records. In our study, we classified species as translocated based on the original data structure from Tedesco (2017) and the spatial scale. When a species was identified as non-native in the data, we considered them translocated if it had also been recorded as native species in other river basins within the same country. This definition at country scale is commonly used in some previous studies (Liu et al. 2017; Vitule et al. 2019; USGS 2023). Following your suggestion, we have expanded our discussions regarding the identification and differentiation of translocated species (lines 305-308):

Lines 305-308: “Identifying translocated non-native species from native species remains a challenging task, primarily due to the limited availability of historical records. Additionally, distinguishing them from alien species is heavily contingent on the chosen spatial scale in this study.”

Additionally, we conducted further analyses by extending the spatial scale from the country perspective to the level of biogeographical realms, following the methodology utilized in previous studies (Villéger et al. 2011; Su et al. 2021). Under this framework, species introduced from different realms were categorized as alien species, while those introduced from different regions within the same realm were considered translocated species. This approach provides a more nuanced understanding of the natural biogeography of non-native species. Through these updated classifications, we established new associations between alien-native phylogenetic distances and whether the alien fishes occur (total n = 126419 records), as well as translocation-native phylogenetic distances and whether the translocated fishes occur (total n = 281649 records). The results of these additional analyses were highly consistent with our previous findings (Fig. S4), further confirming our primary conclusion that non-native species were more likely to succeed when they were phylogenetically closely related to native fishes. We have integrated these supplementary analyses into the Results section (lines 160-163) and Methods section (lines 385-393; lines 485-491). We thank you

sincerely for your insightful comment and the valuable improvement it has brought to our study.

Lines 160-163: “This finding was further confirmed after excluding the non-native species within the two dominant families, Cyprinidae and Salmonidae (Fig. S3), and after redefining the alien and translocated non-native species on the biogeographical realm scale (Fig. S4).”

Lines 385-393: “To more accurately depict the natural distribution of non-native fish species while minimizing the influence of administrative boundaries, we also broadened our analysis from a country-level perspective to a biogeographical realm scale. This allowed us to redefine the categories of alien and translocated fish species, in accordance with previous studies^{1,8}. Specifically, we defined alien species as those originating from different biogeographical realms, and translocated species as those moved within the same realm. We then compared the relationships between the occurrence likelihood of non-native species and nonnative-native phylogenetic relatedness under these two distinct definitions.”

Lines 485-491: “Following the redefinition of alien and translocated non-native species on the biogeographical realm scale, we obtained the corresponding relationship between alien-native phylogenetic distances and whether the alien fishes occur (total $n = 126419$ records), as well as translocation-native phylogenetic distances and whether the translocated fishes occur (total $n = 281649$ records) and re-examined the impact of phylogenetic relatedness on the occurrence likelihood of non-native fish species at the global scale.”

Fig. S4 | Relationships between the probability of non-native fish occurrence and nonnative-native phylogenetic distance after redefining the alien and translocated non-native species on the biogeographical realm scale. **a** Occurrence probability of alien fish species with alien-native MPD. **b** Occurrence probability of alien fish species with alien-native MNTD. **c** Occurrence probability of translocated fish species with translocation-native MPD. **d** Occurrence probability of translocated fish species with translocation-native MNTD. Predictive curves (with 95% confidence intervals) were derived from generalized linear mixed models (GLMMs) assuming a binomial error distribution. Statistical significance (P values), variance explained (R^2_{marginal} for the fixed effect and $R^2_{\text{conditional}}$ for both the fixed and random effects), and sample size (n) are presented in the figure.

Tedesco, P. A. et al. A global database on freshwater fish species occurrence in drainage basins. *Sci. Data* **4**, 2017141 (2017).

Liu, C., He, D., Chen, Y., Olden, J. D. Species invasions threaten the antiquity of China's freshwater fish fauna. *Divers. Distrib.* **23**, 556-566 (2017).

Vitule, J. R. et al. Intra-country introductions unraveling global hotspots of alien fish species. *Biodivers. Conserv.* **28**, 3037-3043 (2019).

U.S. Geological Survey. Nonindigenous Aquatic Species Database. Gainesville, Florida. Accessed [10/30/2023]. (2023).

Villéger, S., Blanchet, S., Beauchard, O., Oberdorff, T., Brosse, S. Homogenization patterns of the world's freshwater fish faunas. *Proc. Natl. Acad. Sci. USA* **108**, 18003-18008 (2011).

Su, G., Logez, M., Xu, J., Tao, S., Villéger, S., Brosse, S. Human impacts on global freshwater fish biodiversity. *Science* **371**, 835-838 (2021).

The discussion of translocation hotspots seems to come out of nowhere (not prominent in the results or study set-up).

Response: We agree with this comment. This paragraph discussing translocation hotspots has been removed, and the essential information has been integrated into the preceding paragraph (lines 199-203):

“Among these, alien fishes have successfully established in nearly 50% of these river basins spanning all the biogeographical realms. In contrast, translocated fishes have colonized approximately 20% of these river basins, with a notable concentration in the Nearctic and Palearctic realms of the northern hemisphere.”

Another recent study (Qian et al. 2022), demonstrated that "Support for Darwin's naturalization and preadaptation hypotheses in freshwater fish assemblages depends on the phylogenetic diversity of assemblages within the invaded watersheds." It may be relevant to cite this work and possibly additional references mentioned therein in this context.

Response: We have cited this excellent work on the phylogenetic diversity of freshwater fish communities in North America (Introduction: lines 100-103; Discussion: lines 294-297).

Lines 100-103: “Furthermore, while taxonomic diversity metrics such as species richness may not adequately capture the diversity of ecological functions they support, comprehensive diversity metrics, such as the phylogenetic diversity of native communities, should be considered simultaneously^{1,33}.”

Lines 294-297: “These results indicate that understanding biotic resistance in fish communities may be more complicated than initially thought, emphasizing the importance of considering different dimensions of diversity when assessing the relationship between biodiversity and biological invasions^{1,33}.”

1. Su, G., Logez, M., Xu, J., Tao, S., Villéger, S., Brosse, S. Human impacts on global freshwater fish biodiversity. *Science* **371**, 835-838 (2021).

33. Qian, H. et al. Effects of non-native species on phylogenetic dispersion of freshwater fish communities in North America. *Divers. Distrib.* **29**, 143-156 (2022).

Finally, while the original data used in this study have already been published, it is important for authors to fully release derived data and indices.

Response: The derived data, indices, and associated R codes will be made fully available on the Figshare repository upon acceptance.

Minor comments:

In 57 what is "their" referring to?

Response: We have rephrased this sentence to make it clearer (lines 53-56):

“Recognizing the global geographical patterns of these non-native species and comprehending the factors behind their prevalence in specific regions is essential for the effective management of these species and forecasting future invasions.”

In 68-9 sentence needs work

Response: We have rephrased this sentence (lines 63-65):

“Nevertheless, there is still a deficiency in distinguishing translocated non-native fishes from alien species and evaluating the difference in their global biogeographical patterns.”

In 75-6 sentence needs work

Response: We have rephrased this sentence (lines 69-71):

“Although these approaches provide valuable insights, they may pose challenges in reaching general conclusions and predictions for non-native fish species.”

In 77 which traits?

Response: We have rephrased this sentence (lines 71-73):

“The challenge primarily arises due to variations in ecological characteristics among species and habitats, coupled with the inherent difficulty in identifying and measuring functional traits^{6,19}.”

In 348-9 Unexpected opening for the discussion not focusing on the results of this study, especially since it was done on the existing data

Response: We have removed this sentence and initiated the discussion by summarizing our main results (lines 189-197):

“Drawing on an updated global biogeography of alien and translocated non-native fish species, our study revealed a notably elevated occurrence probability of non-native fish species when they were closely related to native species, irrespective of at global, biogeographical realm, or country scales. Our findings propose that native fish communities hosting close relatives may be particularly suitable for non-native fishes, providing them with advantages rather than presenting challenges through potential intensified competition. The global patterns observed in non-native freshwater fishes and their phylogenetic associations could establish a basis for comprehending and forecasting future fish invasions in freshwater ecosystems.”

Fig 2 While maps are insightful, it would be good to show univariate relationships for all of the variables of interest as well (e.g., alien richness vs. alien-native MPD, alien % vs. alien-native MPD; same for co-variates). At present it is not possible to tell how much these results (including figure 3) are affected by unequal sample distribution across gradients.

Response: We have supplemented all the univariate relationships following your suggestion (Fig. S8).

Fig. S8 | Assessment of correlation relationships among key predictor variables. For alien fish species: **a** Relationship between native richness and MPD; **b** Relationship between native richness and MNTD; **c** Relationship between native MPD and MPD; **d** Relationship between

native MNTD and MNTD; **e** Relationship between native richness and native MPD; **f** Relationship between native richness and native MNTD; **g** Relationship between alien richness and MPD; **h** Relationship between alien richness and MNTD; **i** Relationship between alien percentage and MPD; and **j** Relationship between alien percentage and MNTD. For translocated fish species: **k** Relationship between native richness and MPD; **l** Relationship between native richness and MNTD; **m** Relationship between native MPD and MPD; **n** Relationship between native MNTD and MNTD; **o** Relationship between native richness and native MPD; **p** Relationship between native richness and native MNTD; **q** Relationship between translocated richness and MPD; **r** Relationship between translocated richness and MNTD; **s** Relationship between translocated percentage and MPD; and **t** Relationship between translocated percentage and MNTD.

Fig 5 is difficult to interpret

Response: The figure illustrates the consistent relationship between the probability of non-native fish occurrence and phylogenetic relatedness within each biogeographical realm. The primary focus is on the negative logistic relationships depicted on the right side. The chord diagrams on the left side serve primarily to distinguish between different realms and to depict the associations between alien fishes and countries for each realm. Additionally, we noted the most frequently introduced alien fishes and the countries responsible for introducing the highest number of alien fishes below the chord diagrams.

We have made additional modifications to the figure legends (the Fig. 5 has been referenced as Fig. 4 in the revised MS):

“Fig. 4 | Relationships between the probability of non-native fish occurrence and nonnative-native phylogenetic distance in six biogeographical realms of the world (a Palearctic, b Afrotropic, c Neotropic, d Australasia, e Indo-Malay, and f Nearctic). For each biogeographical realm, i occurrence probability of alien fish species with alien-native MPD, ii occurrence probability of alien fish species with alien-native MNTD, iii occurrence probability of translocated fish species with translocation-native MPD, and iv occurrence probability of translocated fish species with translocation-native MNTD. Predictive curves were generated from the GLMMs, with solid lines representing significant effects ($P < 0.05$). Statistical significance (P values) and sample size (n) are displayed in the figure. For each realm, a chord diagram illustrates the network relationship between countries and alien fish species, showing which alien fishes have been introduced into specific countries and which countries have introduced specific alien fish species. The most frequently introduced alien fish species and the country that introduced the largest number of alien fish species are noted below the chord diagrams for clarity.”

Fig 6(a) the effect of native fish richness is as strong or stronger than the effect of MPD, why is the arrow thickness not aligned with that effect size? Why does the model not account for covariation between native richness and native MPD?

Response: In Fig. 6a (which has been referenced as Fig. 5 in the revised MS), the effect size for native fish richness is 0.58, while the effect size for MPD is -0.53. We have double-checked and confirmed that the arrow thickness in the figure corresponds to the effect size.

In the structural equation modeling (SEM), the correlation between the exogenous variables, specifically native richness and native MPD in our models, is tested independently. This correction does not affect the results of the regressions (i.e. the path coefficients). As a result, the correlation path between the exogenous variables is often omitted from the final figure if it is not the primary focus of the research (e.g. Li Y et al. 2023; Luo et al. 2023; Weeks et al. 2022).

Additionally, in the Method section, “We verified that the inclusion of both relatedness and diversity metrics did not lead to multicollinearity issues because there were low correlations among them (Fig. S8) and *VIFs* (variance inflation factors) for all metrics remained below three” (lines 521-524). Therefore, we omitted the correlation path in our SEM.

Li, Y. et al. Multitrophic arthropod diversity mediates tree diversity effects on primary productivity. *Nat Ecol Evol* 7, 832-840 (2023).

Luo, S. et al. Higher productivity in forests with mixed mycorrhizal strategies. *Nat. Commun.* 14, 1377 (2023).

Weeks, B. C., Naeem, S., Lasky, J. R., Tobias, J. A. Diversity and extinction risk are inversely related at a global scale. *Ecol. Lett.* 25, 697-707 (2022).

Reviewer #2 (Remarks to the Author):

The manuscript by Xu et al. investigates the influence of phylogenetic relatedness on invasion success of non-native fish in more than 3000 drainages across the world. Additionally, the study investigates differences in alien fish, which originates from outside- and inside a given country (latter defined as translocated species). Lastly, the study also investigates the influence of local diversity on invasion success.

Response: We appreciate your positive evaluation of our manuscript and the constructive comments that have significantly enhanced its quality.

The overall theme definitely relevant, but how novel is it compared to previous publications? The Xu et al. 2022 GCB paper describes the same general finding, but for lakes (not rivers). The addition with a contrast between alien and translocated fish is an interesting addition, but has some caveats (see below).

Response: Yes, our present study, along with our prior publication (Xu et al. 2022), revolves around a common theme: exploring how ecological and evolutionary similarities between non-native and native species influence the invasion success of these non-natives. In our earlier research, which was conducted in Swedish freshwater

lakes, one of our principal findings highlighted that exotic species closely related to resident species were more likely to establish successfully. This discovery provided the initial supporting evidence for Darwin's preadaptation hypothesis within freshwater fish communities. Building upon this foundation, our current study seeks to determine whether this phenomenon is widespread among all non-native fish species across global freshwater ecosystems. If this hypothesis proves to be valid on a global scale, it would offer a practical and convenient means of understanding, predicting, and managing non-native fishes. This overarching motivation underpins our current research.

As we collected data to address this question, we became aware that the global biogeographical patterns of non-native fishes remain poorly understood, with only a handful of studies using limited data (based on 1055 river basins worldwide) to examine them (Leprieur et al. 2008; Villéger et al. 2011). This limited understanding hampers our ability to comprehensively assess the phylogenetically related mechanisms on a global scale. Thus, one additional aim of this study is to provide a comprehensive and up-to-date assessment of the global distribution patterns of non-native freshwater fishes. By combining these updated global patterns of non-native fishes with a novel algorithm, we can evaluate the impact of phylogenetic relatedness on global fish invasions.

Recognizing the pivotal role of intra-country introductions in understanding invasions and the resulting fish homogenization (Vitule et al. 2019), we also differentiated between alien fishes and translocated fishes, assessed their biogeographical patterns, and compared the effects of phylogenetic relatedness on their occurrences. We appreciate your positive evaluation of this aspect and have provided a point-by-point response to your questions on this topic below.

The specific differences between the current study and the previously published GCB paper are detailed as follows:

- (1) **Scale and scope of the study.** In the GCB paper, we investigated 965 intentional introductions of 23 freshwater fish species into 673 lakes in Sweden, involving a total of 38 introduced and native fish species. In the current study, we expanded our research to a global scale, examining 14708 species across 3008 river basins in 121 countries, of which 597 species have been introduced as either alien or translocated species. This extensive global dataset enables us to thoroughly examine the effect of phylogenetic relatedness on fish invasions and draw more generalized conclusions regarding the effectiveness of Darwin's preadaptation hypothesis in freshwater ecosystems. Please also refer to the comparison of the phylogenetic tree used in our two studies below:

Phylogeny for 38 species in GCB paper

Phylogeny for 14708 species in current study

(2) **Algorithms and analytical methods.** In the GCB paper, the clear introduction success or failure allowed us to directly test the relationship between success probability of exotic species and phylogenetic relatedness. In the current study, we used a novel analysis method to establish associations between the occurrence probability of non-native species and phylogenetic relatedness (Fig. S7). This approach was necessitated by the fact that we only had extant occurrence data for non-native species in each river basin. Our method offers a general approach to examine the occurrence probability for any extant non-native species and may serve as a valuable reference for similar studies.

Fig. S7 | Conceptual diagram illustrating the approach to associate nonnative-native phylogenetic distances with non-native fish occurrence. For each alien fish species, its presence in one (or multiple) river basin within a country is defined as a success (1), while its absence in all the other basins within that country is considered as a failure (0). Similarly, for each translocated fish species (*i.e.* recorded as native species in other river basins within a country), its presence in one (or multiple) river basin is denoted as a success (1), and its absence in all the other basins within the country is considered as a failure (0). For each non-native fish species in a country, its phylogenetic distance with all native species in each river basin within the country is calculated, regardless of whether the non-native species occurs in that basin. This approach establishes the corresponding relationship between nonnative-native phylogenetic distance and the likelihood of non-native fish occurrence, enabling the examination of the effect of phylogenetic relatedness on non-native fish occurrence.

(3) **Updating global patterns of non-native fish species.** The updated maps, covering 3119 river basins (Figure 2), represent the most completed assessment of global non-native fish species to date, with approximately three times more river basins than in previous research (Leprieur et al. 2008; Villéger et al. 2011). We believe that the global hotspots identified in this study can provide valuable insights into the biogeography of fish invasion for fish ecologists and assist policy makers in formulating specific management policies aimed at mitigating the negative impacts of non-native fish species.

Fig. 2 | Global geographical distribution of alien and translocated freshwater fish species. **a** Geographical pattern of alien fish richness. **b** Geographical pattern of alien fish percentage. **c** Mean phylogenetic distance (MPD) between alien and native fish species in river basins where alien fish species occur. **d** Nearest phylogenetic distance (MNTD) between alien and native fish species in river basins where alien fish species occur. **e** Geographical pattern of translocated fish richness. **f** Geographical pattern of translocated fish percentage. **g** Mean phylogenetic distance (MPD) between translocated and native fish species in river basins where translocated fish species occur. **h** Nearest phylogenetic distance (MNTD) between translocated and native fish species in river basins where translocated fish species occur. The percentage represents the ratio of non-native species richness to the total species richness in each river basin. The number of river basins used for assessing these patterns is displayed at the top of each panel.

(4) **Distinguishing translocated species from alien species.** In this study, we made a clear distinction between translocated fishes and alien fishes. We presented their global distribution patterns separately at both the river basin scale (Fig. 2) and the country scale (Fig. S2). Moreover, we illustrated the consistent effects of phylogenetic relatedness on their global occurrences.

Fig. S2 | Geographical distribution of alien and translocated freshwater fish species across different countries. **a** Geographical pattern of alien fish richness. **b** Geographical pattern of translocated fish richness. **c** Geographical pattern of alien fish percentage. **d** Geographical pattern of translocated fish percentage. The percentage is referred to the ratio of non-native species richness to the total species richness in each country. The number of countries used for assessing these patterns is shown at the top of each panel.

Xu, M. et al. Exotic fishes that are phylogenetically close but functionally distant to native fishes are more likely to establish. *Global Change Biol.* **28**, 5683-5694 (2022).

Leprieur, F., Beauchard, O., Blanchet, S., Oberdorff, T., Brosse, S. Fish invasions in the world's river systems: when natural processes are blurred by human activities. *PLoS Biol.* **6**, e28 (2008).

Villéger, S., Blanchet, S., Beauchard, O., Oberdorff, T., Brosse, S. Homogenization patterns of the world's freshwater fish faunas. *Proc. Natl. Acad. Sci. USA* **108**, 18003-18008 (2011).

Vitule, J. R. et al. Intra-country introductions unraveling global hotspots of alien fish species. *Biodivers. Conserv.* **28**, 3037-3043 (2019).

Including fish species translocated between drainages within a country is definitely a strength of this study. However, it is a bit problematic that movement of fish species within and between countries are used as defining translocated- and alien fishes, respectively. Countries are made by political borders and translocations within countries may be between drainages with very different colonization history, whereas translocation between countries may be within the same major drainages. This is further problematic, since not all countries are of the same size, which will influence the use of the terms for different regions. This may very well explain the countries with the highest amount of translocated fish species are also among the largest in the world.

Response: Thank you for this valuable comment. The definition of the alien and translocated species in the context of non-native species invasions is indeed a matter of debate. Various criteria, such as political borders and natural biogeography, have been

employed to categorize these species. Typically, alien species are defined as those introduced from other countries, while translocated species refer to those introduced from other regions within a country (Liu et al. 2017; Vitule et al. 2019; USGS 2023). However, this distinction can be expanded to a continental level, designating species imported from other continents as alien species and those introduced from other regions within a continent as translocated species (Anas et al. 2021; Villéger et al. 2014). Furthermore, some global studies have used biogeographical realms (i.e. Afrotropical, Australasia, Indo-Malay, Nearctic, Neotropic, Oceania and Palearctic) to categorize the alien and translocated species. In this context, species introduced from other realms are considered alien species, while those introduced from other regions within a realm are classified as translocated species (Villéger et al. 2011; Su et al. 2021). This approach may better reflect the natural biogeography of non-native species.

To address this issue, following the previous studies (Villéger et al. 2011; Su et al. 2021), we redefined alien and translocated fish species at the scale of biogeographical realm. This redefinition not only changes the ratio of alien to translocated species in each river basin, but also alters the number of river basins where a specific non-native species is present or absent (previously defined within a country, now within a realm). Subsequently, we established the updated corresponding relationship between alien-native phylogenetic distances and whether the alien fishes occur (total $n = 126419$ records), as well as translocation-native phylogenetic distances and whether the translocated fishes occur (total $n = 281649$ records). We retested the effect of phylogenetic relatedness on the occurrence probability of non-native fish species and obtained highly consistent patterns with our previous analyses (Fig. S4). This result validates our main finding that non-native species are more likely to succeed when they are closely related to native fishes. We have included this supplementary information in both the Results section (lines 160-163) and Methods section (lines 385-393; 485-491). Additionally, we have also expanded our discussions regarding the identification and differentiation of translocated species (lines 305-308). We thank you sincerely for your insightful comment and the valuable improvement it has brought to our study.

Lines 160-163: “This finding was further confirmed after excluding the non-native species within the two dominant families, Cyprinidae and Salmonidae (Fig. S3), and after redefining the alien and translocated non-native species on the biogeographical realm scale (Fig. S4).”

Lines 385-393: “To more accurately depict the natural distribution of non-native fish species while minimizing the influence of administrative boundaries, we also broadened our analysis from a country-level perspective to a biogeographical realm scale. This allowed us to redefine the categories of alien and translocated fish species, in accordance with previous studies^{1,8}. Specifically, we defined alien species as those originating from different biogeographical realms, and translocated species as those moved within the same realm. We then compared the relationships between the occurrence likelihood of non-native species and nonnative-native phylogenetic relatedness under these two distinct definitions.”

Lines 485-491: “Following the redefinition of alien and translocated non-native species on the biogeographical realm scale, we obtained the corresponding relationship between alien-native phylogenetic distances and whether the alien fishes occur (total $n = 126419$ records), as well as translocation-native phylogenetic distances and whether the translocated fishes occur (total $n = 281649$ records) and re-examined the impact of phylogenetic relatedness on the occurrence likelihood of non-native fish species at the global scale.”

Lines 305-308: “Identifying translocated non-native species from native species remains a challenging task, primarily due to the limited availability of historical records. Additionally, distinguishing them from alien species is heavily contingent on the chosen spatial scale in this study.”

Fig. S4 | Relationships between the probability of non-native fish occurrence and nonnative-native phylogenetic distance after redefining the alien and translocated non-native species on the biogeographical realm scale. a Occurrence probability of alien fish species with alien-native MPD. **b** Occurrence probability of alien fish species with alien-native MNTD. **c** Occurrence probability of translocated fish species with translocation-native MPD. **d** Occurrence probability of translocated fish species with translocation-native MNTD. Predictive curves (with 95% confidence intervals) were derived from generalized linear mixed models (GLMMs) assuming a binomial error distribution. Statistical significance (P values), variance explained (R^2_{marginal} for the fixed effect and $R^2_{\text{conditional}}$ for both the fixed and random effects), and sample size (n) are presented in the figure.

Liu, C., He, D., Chen, Y., Olden, J. D. Species invasions threaten the antiquity of China's freshwater fish fauna. *Divers. Distrib.* **23**, 556-566 (2017).

Vitule, J. R. et al. Intra-country introductions unraveling global hotspots of alien fish species. *Biodivers. Conserv.* **28**, 3037-3043 (2019).

U.S. Geological Survey. Nonindigenous Aquatic Species Database. Gainesville, Florida. Accessed [10/30/2023]. (2023).

Anas, M. U. M., Mandrak, N. E. Drivers of native and non-native freshwater fish richness across North America: Disentangling the roles of environmental, historical and anthropogenic factors. *Global Ecol. Biogeogr.* **30**, 1232-1244 (2021).

Villéger, S., Grenouillet, G., Brosse, S. Functional homogenization exceeds taxonomic homogenization among European fish assemblages. *Global Ecol. Biogeogr.* **23**, 1450-1460 (2014).

Villéger, S., Blanchet, S., Beauchard, O., Oberdorff, T., Brosse, S. Homogenization patterns of the world's freshwater fish faunas. *Proc. Natl. Acad. Sci. USA* **108**, 18003-18008 (2011).

Su, G., Logez, M., Xu, J., Tao, S., Villéger, S., Brosse, S. Human impacts on global freshwater fish biodiversity. *Science* **371**, 835-838 (2021).

I can understand how MPD and MNTD is calculated for each non-native fish species and also for each of the native species present in a drainage. However, I cannot understand how this can also be used as a measure of diversity of native fish species (L193-195)?

Response: To quantify the phylogenetic diversity of native fish species, we utilized two commonly employed metrics, MPD (mean phylogenetic distance) and MNTD (mean nearest taxon distance). These metrics are widely used in ecological studies to assess the phylogenetic diversity of a community (Swenson 2014; Tucker et al. 2017).

Specifically, MPD was calculated by mean pairwise phylogenetic distances among all the native species in a river (i.e. summing all pairwise phylogenetic distances among the native species in a river basin and then dividing this sum by the total number of native species present in that basin) (Webb et al. 2002). In essence, native MPD provides an average measure of how different or similar the native species are in terms of their evolutionary relationships.

MNTD, on the other hand, was computed by mean phylogenetic distance of each native species to its nearest native neighbor in a basin (i.e. summing the minimum phylogenetic distance of each native species and then dividing this sum by the total number of native species present in that basin) (Webb et al. 2002). MNTD provides insights into the phylogenetic relatedness between each native species and its closest relatives in the community.

To make it clear, we have added specific descriptions on these two metrics in the Methods section (lines 419-423):

“For each river basin, we further assessed the phylogenetic diversity of native species by employing two additional metrics with the *picante* R package⁵⁵: native MPD (calculated by mean pairwise phylogenetic distances among all the native species in a river) and native MNTD (computed by mean phylogenetic distance of each native species to its nearest native neighbor in a basin)^{56,57}.”

Swenson, N. G. Functional and phylogenetic ecology in R. Springer (2014).

Tucker, C. M. et al. A guide to phylogenetic metrics for conservation, community ecology and macroecology. *Biol. Rev.* 92, 698-715 (2017).

Webb, C. O., Ackerly, D. D., McPeck, M. A., Donoghue, M. J. Phylogenies and community ecology. *Annu. Rev. Ecol. Syst.* 33, 475-505 (2002).

I am also questioning how much of the results that are driven by some not do novel patterns, given that carp and rainbow trout are the most frequent alien species. It is not surprising that rainbow trout would mainly be able to establish populations in cold-water rivers, which are already the habitat of native salmonids, whereas carp would establish populations in warm, slow-flowing rivers, which are home to native cyprinids. It is then not really about ecological niche, but rather about general environmental requirements of these two important families. It would be good, if the authors could estimate how much of the effect that is caused by such e.g. cyprinid-salmonid environmental preferences.

Response: Thank you for your valuable suggestion. In this study, we consider the shared environmental requirements of non-native fishes and their closely related natives as an important mechanism for the patterns (supporting preadaptation hypothesis). We found that both types of non-native fishes are more likely to succeed when they are closely related to native fishes. However, as you said, Cyprinidae and Salmonidae include the most common non-native fish species and the environmental preferences of these key non-native species may have a substantial impact on our current findings. This potentially limits the generalizability of our findings. To address this concern, we conducted further analyses following your guidance. Specifically, we excluded data involving non-native species from these two families and re-evaluated the relationship between occurrence probability and phylogenetic relatedness. Remarkably, we found that non-native species still exhibited a higher likelihood of success when they were phylogenetically closely related to native fishes (Fig. S3). Furthermore, as you expertly demonstrated, we also observed similar patterns when only examining non-natives from these two families. These consistent results reinforce our confidence in the existence of a pervasive higher success for non-native fish species when they are closely related to native species. We have incorporated these additional analyses into both the Results section (lines 160-161) and Methods sections (477-484), and we greatly appreciate your insightful comment.

Lines 160-161: “This finding was further confirmed after excluding the non-native species within the two dominant families, Cyprinidae and Salmonidae (Fig. S3)”

Lines 477-484: “Considering that Cyprinidae and Salmonidae include the most common non-native fish species, and with the former showing a preference for establishing populations in warm rivers while the latter primarily thrive in cold rivers, the environmental preferences of these key non-native species may have a substantial impact on our current findings, potentially limiting their generalizability. To address this concern, we conducted additional analyses by excluding data related to non-native species within these two dominant families. Subsequently, we re-evaluated the relationship between the occurrence probability and phylogenetic relatedness across the globe.”

Fig. S3 | Relationships between the probability of non-native fish occurrence and nonnative-native phylogenetic distance after excluding the non-native species within the two dominant families, Cyprinidae and Salmonidae. a Occurrence probability of alien fish species with alien-native MPD. **b** Occurrence probability of alien fish species with alien-native MNTD. **c** Occurrence probability of translocated fish species with translocation-native MPD. **d** Occurrence probability of translocated fish species with translocation-native MNTD. Predictive curves (with 95% confidence intervals) were derived from generalized linear mixed models (GLMMs) assuming a binomial error distribution. Statistical significance (P values), variance explained (R^2_{marginal} for the fixed effect and $R^2_{\text{conditional}}$ for both the fixed and random effects), and sample size (n) are presented in the figure.

Regarding the observed positive species diversity-occurrence relationship, I think that it is worth mentioning that it can also be explained by island biogeography (distance to the mainland) theory: if drainages are generally more isolated, they would tend to have more depauperate native diversity (e.g. colonization since last glaciation). Such more isolated drainages would potentially also tend to have less non-native species. But even drainage size could also influence this pattern: large drainages tend to have more native species, but also have more points of entry for alien species.

Response: We have extended our discussions regarding the positive species diversity-occurrence relationship in the context of the theory of island biogeography, as per your suggestion (lines 286-290):

“However, it is worth noting that the positive species diversity-occurrence relationship may also be attributed to the isolation and size of the river basin, as predicted by the theory of island biogeography⁴⁴. While more isolated and small river basins tend to have lower native diversity, they may also support fewer non-native species. This aspect warrants further investigation in future studies.”

Reviewer #1 (Remarks to the Author):

For the most part, the authors have thoroughly addressed reviewer concerns. I have the following remaining comments:

1. For a ms focusing on invasive species, the inconsistent use of terminology is not appropriate. "Alien" is not a scientific term, and has, fortunately, been declining in the literature. Please use "non-native" (which you already do but inconsistently) to indicate origins and "invasive" for non-native species which spread/are reproducing (e.g., Havel et al. 2015). NB figure inserts, should also say "non-native".
2. One of my two key comments was only partially addressed by considering biogeographical provinces. To re-iterate, countries (or biogeographical provinces are not river basins, and even if you focused on basins, only a few basins have long-term geological features blocking migrations). Please discuss this limitation more fully, especially since the second reviewer had the same concern.
3. Fig. S8. a) Data of this density really need to be plotted using a 3D space such as a heatmap, or at the very least using smaller, transparent (higher alpha), unfilled symbols.
b) Allen is misspelled (and should be non-native, see above)
c) Many relationships are notably non-linear, native richness covers a truncated gradient (data are not distributed evenly across the richness gradient) or are driven by very few extreme outliers. The readers need to know the relative effect of outliers, and these limitations of data distribution should be more clearly discussed.
4. "translocated fishes have colonized approximately 20% of these river basins, with a notable concentration in the Nearctic and Palearctic realms of the northern hemisphere" - where most of the sampling effort was concentrated. Perhaps worth discussing a bit more.

Reviewer #2 (Remarks to the Author):

General comments to authors

The authors have well addressed my previous comments in their reply and made changes to the manuscript accordingly. I therefore only have one major comment, which relates to the minor comments, at this point and would like to congratulate the authors on a very interesting paper, which I am sure will be received well in the scientific community.

One major comment

In many places in the results and in the discussion, the text suggest that it is the presence of closely related species, which facilitates the success of non-natives. However, it seems much more plausible that the presence of close relatives is an indicator of habitat suitability, which can explain the correlation between presence of close relatives and establishment success. This does not rule out a negative influence of competition, but just suggest that a negative effect of competition from close relatives is smaller than the positive effects of the habitat suitability, which the close relatives indicate. I strongly suggest that the authors clarify this throughout the text.

Specific minor comments

L92-94: It is not the high native diversity per se, which is facilitating the establishment success of non-native species. But a high native diversity would often be a result of an environment that allows co-existence of multiple species (e.g. through high niche diversity and habitat size) and thereby also facilitate establishment success of non-native species. Just be careful about the wording, as the first and second part of the sentence is actually not saying the same thing. Maybe exchange "facilitate" with "predict".

L175-178: Also here, it does not appear to be a logic conclusion that higher native species diversity would indirectly promote non-native success by reducing phylogenetic distance. Also this would more likely both to be promoted by the environment (both higher native phylogenetic diversity and higher non-native success). That high phylogenetic diversity is a strong predictor does not imply causation! It is the same problem, when referring to "effect" rather than "predict".

L192-195: See major comment above.

L237-238: See major comment above.

Response to reviewer's comments

Reviewer #1 (Remarks to the Author):

For the most part, the authors have thoroughly addressed reviewer concerns. I have the following remaining comments:

Response: Thank you for your positive evaluation of our revised manuscript. Please find below our point-by-point response to the remaining comments you provided.

1. For a ms focusing on invasive species, the inconsistent use of terminology is not appropriate. "Alien" is not a scientific term, and has, fortunately, been declining in the literature. Please use "non-native" (which you already do but inconsistently) to indicate origins and "invasive" for non-native species which spread/are reproducing (e.g., Havel et al. 2015). NB figure inserts, should also say "non-native".

Response: Traditionally, in comparison to native species, we concur that using the term “non-native” is more suitable to denote the origin. However, in alignment with recent studies (Villéger et al. 2011; Liu et al. 2017; Vitule et al. 2019; Anas et al. 2021; Su et al. 2021), we made a distinction between two types of non-native species: those introduced from other realms (referred to as alien species) and those translocated within a realm but across different river basins (referred to as translocated species). We then proceeded to compare the impact of phylogenetic relatedness on the occurrences of these two types of non-native species. Considering both types fall under the umbrella of “non-native”, using “non-native” instead of “alien” might lead to ambiguity in distinguishing between alien and translocated species. Following conventions in some freshwater fish invasion studies (Villéger et al. 2011; Su et al. 2021; Xu et al. 2022), we have opted to use the more formal term “exotic” to replace “alien” throughout the manuscript, while retaining “non-native” to encompass both types comprehensively.

Villéger, S., Blanchet, S., Beauchard, O., Oberdorff, T., Brosse, S. Homogenization patterns of the world's freshwater fish faunas. *Proc. Natl. Acad. Sci. USA* **108**, 18003-18008 (2011).

Liu, C., He, D., Chen, Y., Olden, J. D. Species invasions threaten the antiquity of China's freshwater fish fauna. *Divers. Distrib.* **23**, 556-566 (2017).

Vitule, J. R. et al. Intra-country introductions unraveling global hotspots of alien fish species. *Biodivers. Conserv.* **28**, 3037-3043 (2019).

Anas, M. U. M., Mandrak, N. E. Drivers of native and non-native freshwater fish richness across North America: Disentangling the roles of environmental, historical and anthropogenic factors. *Global Ecol. Biogeogr.* **30**, 1232-1244 (2021).

Su, G., Logez, M., Xu, J., Tao, S., Villéger, S., Brosse, S. Human impacts on global freshwater fish biodiversity. *Science* **371**, 835-838 (2021).

Xu, M. et al. Exotic fishes that are phylogenetically close but functionally distant to native fishes are more likely to establish. *Global Change Biol.* **28**, 5683-5694 (2022).

2. One of my two key comments was only partially addressed by considering biogeographical provinces. To re-iterate, countries (or biogeographical provinces are not river basins, and even if you focused on basins, only a few basins have long-term geological features blocking migrations). Please discuss this limitation more fully, especially since the second reviewer had the same concern.

Response: We have extended our discussions to emphasize this limitation more comprehensively (lines 307-323):

“Firstly, identifying translocated non-native species from native ones remains a formidable challenge. Determining definitively whether a species has been translocated among river basins or has a historical presence in a particular basin is often impossible, primarily due to the extensive connectivity among basins and the scarcity of historical records. In this study, direct observations of translocated fish species were not conducted. Instead, we classified a species as translocated when it was identified as non-native in a specific river basin while simultaneously being recorded as a native species in other river basins within the same country. The absence of clear records for translocated species may introduce bias into the geographical patterns we identified, potentially influencing our analysis of the underlying drivers. For instance, our findings indicated a predominant occurrence of fish translocations in Nearctic and Palearctic realms of the northern hemisphere. However, it is plausible that the majority of the sampling effort was concentrated in these regions, implying a potential bias in our current results. While an increasing number of studies recognized the importance of identifying translocated species^{1,8,13,14}, it is crucial for future research to prioritize the explicit and extensive documentations of species translocation among different river basins.”

3. Fig. S8. a) Data of this density really need to be plotted using a 3D space such as a heatmap, or at the very least using smaller, transparent (higher alpha), unfilled symbols.

b) Allen is misspelled (and should be non-native, see above)

c) Many relationships are notably non-linear, native richness covers a truncated gradient (data are not distributed evenly across the richness gradient) or are driven by very few extreme outliers. The readers need to know the relative effect of outliers, and these limitations of data distribution should be more clearly discussed.

Response:

a) We have updated this figure using smaller and more transparent points.

b) We have substituted the term “Alien” with “Exotic”. When both alien and translocated species are considered as non-native species, using “non-native” in replace of “alien” poses challenges in distinguishing and comparing these two types of non-native species. Please also refer to our response to question 1 for further clarification.

c) We have added additional discussions on the limitations associated with data distribution and extreme outliers, as per your suggestion (lines 339-341):

“Finally, it's worth noting that our comprehensive and intricate database may exhibit uneven data distributions and potential extreme outliers, which could have also influenced our results.”

4. "translocated fishes have colonized approximately 20% of these river basins, with a notable concentration in the Nearctic and Palearctic realms of the northern hemisphere" - where most of the sampling effort was concentrated. Perhaps worth discussing a bit more.

Response: We have added further discussions on this particular point (lines 317-320):

“For instance, our findings indicated a predominant occurrence of fish translocations in Nearctic and Palearctic realms of the northern hemisphere. However, it is plausible that the majority of the sampling effort was concentrated in these regions, implying a potential bias in our current results.”

Reviewer #2 (Remarks to the Author):

General comments to authors

The authors have well addressed my previous comments in their reply and made changes to the manuscript accordingly. I therefore only have one major comment, which relates to the minor comments, at this point and would like to congratulate the authors on a very interesting paper, which I am sure will be received well in the scientific community.

Response: Thank you for your positive evaluation.

One major comment

In many places in the results and in the discussion, the text suggest that it is the presence of closely related species, which facilitates the success of non-natives. However, it seems much more plausible that the presence of close relatives is an indicator of habitat suitability, which can explain the correlation between presence of close relatives and establishment success. This does not rule out a negative influence of competition, but just suggest that a negative effect of competition from close relatives is smaller than the positive effects of the habitat suitability, which the close relatives indicate. I strongly suggest that the authors clarify this throughout the text.

Response: Thank you sincerely for your insightful comments. We have already revised these expressions throughout the text.

Specific minor comments

L92-94: It is not the high native diversity per se, which is facilitating the establishment success of non-native species. But a high native diversity would often be a result of an environment that allows co-existence of multiple species (e.g. through high niche diversity and habitat size) and thereby also facilitate establishment success of non-

native species. Just be careful about the wording, as the first and second part of the sentence is actually not saying the same thing. Maybe exchange “facilitate” with “predict”.

Response: We have replaced the “facilitate” with “predict”.

L175-178: Also here, it does not appear to be a logic conclusion that higher native species diversity would indirectly promote non-native success by reducing phylogenetic distance. Also this would more likely both to be promoted by the environment (both higher native phylogenetic diversity and higher non-native success). That high phylogenetic diversity is a strong predictor does not imply causation! It is the same problem, when referring to “effect” rather than “predict”.

Response: We have revised this sentence (lines 178-181):

“Moreover, it is worth noting that higher levels of species richness appeared to be indirectly linked to the non-native species occurrence by predicting a reduction in the phylogenetic distance between non-native and native species (Fig. S6).”

L192-195: See major comment above.

Response: We have revised this sentence (lines 195-199):

“Our findings suggest that native fish communities hosting close relatives may be particularly favorable for non-native fishes. The habitat adaptation advantages they offer outweigh the potential negative impacts from intensified competition, ultimately promoting the establishment of non-native fish species.”

L237-238: See major comment above.

Response: We have revised this sentence (lines 239-241):

“Collectively, our findings strongly support the pre-adaptation hypothesis, indicating that the presence of close relatives predicts the occurrence of non-native fish species in freshwater ecosystems.”